# Does Awareness of Malaysian Healthy Plate Associate with Adequate Fruit and Vegetable Intake among Malaysian Adults with Non-Communicable Diseases?

**DOI:** 10.3390/nu15245043

**Published:** 2023-12-08

**Authors:** Lay Kim Tan, En Hong Chua, Sumarni Mohd Ghazali, Yong Kang Cheah, Vivek Jason Jayaraj, Chee Cheong Kee

**Affiliations:** 1Sector for Biostatistics & Data Repository, Office of NIH Manager, National Institutes of Health, Ministry of Health Malaysia, Shah Alam 40170, Selangor, Malaysia; vivekjason.j@moh.gov.my (V.J.J.); kee@moh.gov.my (C.C.K.); 2Department of Biomedical Sciences, Faculty of Medicine and Health Sciences, Universiti Putra Malaysia, Serdang 43400, Selangor, Malaysia; ehchua3003@gmail.com; 3Biomedical Epidemiology Unit, Special Resource Centre, Institute for Medical Research, Ministry of Health Malaysia, Shah Alam 40170, Selangor, Malaysia; sumarni.mg@moh.gov.my; 4School of Economics, Finance and Banking, College of Business, Universiti Utara Malaysia, Sintok 06010, Kedah, Malaysia; yong@uum.edu.my

**Keywords:** Malaysian healthy plate, adequate fruit and vegetable intake, NCDs, morbid, Malaysian adults

## Abstract

The healthy eating plate concept has been introduced in many countries, including Malaysia, as a visual guide for the public to eat healthily. The relationship between Malaysian Healthy Plate (MHP) and adequate fruit and vegetable (FV) intake among morbid Malaysian adults is unknown. Hence, we investigated the relationship between awareness of the MHP and FV intake among morbid Malaysian adults. National survey data on 9760 morbid Malaysian adults aged 18 years and above were analyzed. The relationship between awareness of MHP and FV intake among Malaysian adults with obesity, diabetes mellitus, hypertension, and hypercholesterolemia were determined using multivariable logistic regression controlling for sociodemographic characteristics and lifestyle risk factors. Our data demonstrated that MHP awareness is associated with adequate FV intake among the Malaysian adults with abdominal obesity (adjusted odds ratio (aOR): 1.86, 95% confidence interval (CI): 1.05–3.29), diabetes mellitus (aOR: 4.88, 95% CI: 2.13–22.18), hypertension (aOR: 4.39, 95% CI: 1.96–9.83), and hypercholesterolemia (aOR: 4.16, 95% CI: 1.48–11.72). Our findings indicated the necessity for ongoing efforts by policymakers, healthcare professionals, and nutrition educators to promote the concept of MHP and ensure that morbid Malaysian adults consume a sufficient intake of FV or adopt a healthy eating pattern to achieve and maintain optimal health.

## 1. Introduction

The Malaysian Healthy Plate (MHP) concept, which espouses the tagline “*Suku suku separuh*” (Quarter Quarter Half), is a visual guide introduced by the Ministry of Health (MOH) Malaysia in 2016 [1]. The MHP translates the recommendations of the Malaysian Dietary Guidelines (MDG) and the Malaysian Food Pyramid into a ten-inch round plate that is divided into three portions, i.e., two quarters and a half, as a visual aid to Malaysians to practice healthy eating habits [1]. The divided round plate serves as a blueprint of the total food in each food group that needs to be consumed in a meal to achieve a healthy and balanced diet. A healthy plate is comprised of a quarter plate of grains (i.e., rice, other cereals, whole grains, cereal-based products, and tubers), followed by a quarter plate of protein (i.e., fish, poultry, eggs, meat, and legumes) and a half plate of fruit and vegetables (FV) (Figure 1) [1,2]. 

Although the World Health Organization (WHO) recommended a minimum intake of 400 g or five servings of adequate FV per day in the year 2000 [3], there are a number of countries that set their own recommendations for adequate FV intake in their local dietary guidelines. For example, a daily intake of 1.5–2 cup equivalents of fruit and 2–3 cup equivalents of vegetables is recommended by the 2020–2025 Dietary Guidelines for Americans [4]. While the 2016 Dietary Guidelines for China recommend a daily intake of 300–500 g of vegetables and 200–350 g of fruit (equivalent to 4–5 servings of fruit and vegetables) [5], the 2000 Dietary Guidelines for Japan recommend 5–6 servings of vegetables and 2 servings of fruit [6]. In Malaysia, the 2010 Malaysian Dietary Guidelines (MDG) recommend a daily intake of at least five servings of various fruits and vegetables (at least three servings of vegetables and two servings of fruit) for healthy adults [2].

The MHP concept is considered one of the practical approaches aimed at educating and helping the public to practice adequate daily FV intake, as well as the distribution of macro and micronutrients in the diet and calorie restriction in a healthy way [7]. It focuses on the reduction of high portions of staple foods and protein and consequently maximizes the FV in a main meal. Maximization of FV intake in a meal needs to be emphasized as the combination of micronutrients, antioxidants, phytochemicals, and fiber is beneficial for their protective effects against non-communicable diseases (NCDs), including obesity, diabetes mellitus, hypertension, and hypercholesterolemia [2,8]. For example, the high content of beta-carotene, vitamins C, D, and E, polyphenols, minerals, plant sterols, and stanols in FV are not only vital for sustaining bodily functions but also improve glycemic, blood pressure, and blood lipid control [7,9,10].

A diet with a high content of FV within a meal is believed to be able to prevent diabetes mellitus, hypertension, and hypercholesterolemia and findings from several observational studies are in line with this claim. For instance, a large prospective case–control study conducted in eight disparate European populations reported an association between higher FV intake with lower incidence of type 2 diabetes, which further suggests that diets with even modestly high FV could help prevent the development of type 2 diabetes [11]. The study showed that high FV intake, as indicated by levels of plasma vitamin C, total carotenoids, and individual carotenoid biomarkers, were significantly associated with a lower risk of type 2 diabetes. Recently, Tan and colleagues reported on the relationship between adequate FV intake and cardiovascular risk factors, including diabetes, hypertension, and hypercholesterolemia, using survey data from the 2015 National Health and Morbidity Survey (NHMS) [12]. The study focused on Malaysian adults who were unaware of having diabetes, hypertension, and hypercholesterolemia. The findings demonstrated that adequate vegetable intake alone was associated with a reduced risk of undiagnosed hypertension and hypercholesterolemia. Findings from three prospective cohort studies of 133,468 United States men and women (between the years 1986 and 2010) reported that an increased intake of fruits was inversely associated with 4-year weight change: total fruits −0.53 lb per daily serving (95% CI: −0.61, −0.44), and an increased intake of several vegetables was also inversely associated with weight change: total vegetables −0.25 lb per daily serving (95% CI: −0.35, −0.14) [12].

In the latest dietary guideline for Malaysians (MDG 2020), the food pyramid has been revamped with FV placed at the base of the pyramid, and the recommendation is to have more than two servings of fruits and three servings of vegetables daily to achieve an adequate intake of FV [2]. Lack of awareness of the MHP may be one of the reasons for the low adherence to the recommendations and subsequently low intake of FV. In the National Health and Morbidity Survey (NHMS) 2019, a nationwide community-based cross-sectional study, 79.6% of the general Malaysian adult population had never heard of MHP, 94.9% did not consume adequate amounts of fruit and/or vegetables [13,14], and those who were not aware of MHP were more likely to have inadequate FV intake [13].

Non-pharmacological management (dietary modification) is one of the key recommendations in the Malaysian Clinical Practice Guidelines for the management of type 2 diabetes, hypertension, and dyslipidemia [15,16,17]. Patients with these conditions are advised to adhere to meal plans that meet their individual calorie goals and with a macronutrient distribution that is consistent with a healthy eating pattern, which emphasizes maximization of FV intake for long-term achievement of normoglycemia, lowering blood pressure, lipid control, and ideal weight goals. To date, there is a lack of data on the relationship between awareness of the MHP concept and adequate FV intake among people with morbidities. Hence, in this study, we investigated the association between MHP awareness and adequate FV intake among Malaysian adults with obesity, diabetes mellitus, hypertension, and/or hypercholesterolemia. Data on the relationship between awareness of the MHP concept and adequate FV intake among morbid Malaysian adults are important to provide more insight to public health authorities about the effectiveness of the MHP campaign in this population. The findings from this study may aid the public health authorities in reinforcing the existing strategies to increase and promote awareness and adherence to the MHP practice (behavioral intervention) that focuses on a high-quality, healthy, and balanced diet, which is of paramount importance in the management of morbid Malaysian adults.

## 2. Materials and Methods

### 2.1. Study Design and Sampling

Data from the National Health and Morbidity Survey (NHMS) 2019 were analyzed. Briefly, the NHMS 2019 was a population-based cross-sectional study conducted by the Ministry of Health (MOH) Malaysia. The NHMS 2019 aimed to provide community-based data to determine the prevalence of NCDs, the prevalence of risk factors of NCDs, and relevant critical areas of concern in Malaysia. The collected Malaysian population health information was important to inform the MOH about public health prioritization and programs and to evaluate their impacts while strategizing for future resource allocation [14]. In this present study, we performed a case–reference analytic study, where the case referred to the status of adequate daily FV intake, whilst the reference was inadequate daily FV intake.

The respondents for NHMS 2019 were selected via a two-stage proportionate-to-size cluster sampling design, i.e., the primary stratum and secondary stratum, to ensure a representative sample of the Malaysian population. The primary and secondary sampling units were enumeration blocks (EBs) and living quarters (LQs), respectively, provided by the Department of Statistics of Malaysia. EBs are artificial geographical constructs and are classified as either urban or rural. Urban EBs are gazetted areas with populations of 10,000 individuals or more, whereas rural EBs have populations of less than 10,000. There are more than 75,000 EBs in total, each EB containing between 80 and 120 living quarters (LQs), and each LQ between 500 and 600 individuals. The primary stratum encompassed all the thirteen states and three federal territories in Malaysia, whereas the secondary stratum included the urban and rural areas within the primary stratum. The second stage involved the sampling of LQs from the selected EBs. A total of 475 EBs were selected for NHMS 2019 (362 and 113 urban and rural EBs, respectively. Of these 475 EBs, a total of 5676 LQs were sampled, and 12 LQs were chosen randomly from each EB. The members of all households within the selected LQs who were eligible to participate in the survey were invited to participate in the study. The detailed methodology and sampling design of NHMS 2019 have been reported elsewhere [14].

There was a total of 14,965 respondents for the NCD component of NHMS 2019 [14]. Individuals who were non-Malaysians/citizenship unknown (*n* = 901) and aged below 18 years (*n* = 4304) were excluded from further analyses. After screening for Malaysian adults aged 18 years and above with NCDs (general obesity, abdominal obesity, diabetes mellitus, hypertension, and hypercholesterolemia), the sample was reduced to 9760 individuals. Respondents with diabetes mellitus, hypertension, and hypercholesterolemia were individuals who self-reported being diagnosed with any of these respective chronic diseases by a medical doctor. Out of the 9760 respondents, 264 had missing data on awareness of MHP, and 1332 had missing data on FV intake, and thus were excluded from the analyses, resulting in a total of 8164 respondents included in the logistic regression analysis.

### 2.2. Ethical Consideration

Ethical approval for the NHMS 2019 was granted by the Medical Research and Ethics Committee of the Ministry of Health Malaysia (NMRR-18-3085-44207). All eligible participants were informed about the survey and informed written consent was obtained from all potential respondents prior to conducting the survey interview and related assessments.

### 2.3. Survey Materials and Data Collection

The detailed survey materials and data collection of NHMS 2019 have been published elsewhere [14]. The survey study was conducted between 14 July and 2 October 2019. The data were collected using structured questionnaires that were either interviewed face-to-face by data collectors using an application programmed on tablets or completed by the respondents themselves, who received the questionnaires on the tablet or in paper form.

At enrolment, trained nurses conducted clinical assessments of each respondent, which consisted of anthropometric measurements (i.e., weight, height, length, and waist circumference) [18]. Weight and height were measured using validated and calibrated scales (Tanita personal scale HD 319 and SECA stadiometer 213, respectively). When performed on-site, standard weights were used for calibration. Dietary variables, i.e., awareness of MHP and FV intake, were assessed by a number of items in the questionnaire as described in the following section [14].

### 2.4. Study Variables

#### 2.4.1. Independent Variable

Awareness of the MHP concept was assessed by the question “Have you ever heard of the ‘Malaysian Healthy Plate’/‘Quarter Quarter Half?’”, which had a binary response option (“Yes” or “No”).

#### 2.4.2. Dependent Variable

FV intake of the respondents was assessed using the following four questions: Q1: “In a typical week, how many days did you consume fruits?”; Q2: “Usually on the day that you eat fruits (e.g., apple, orange, banana and so on), how many servings did you take?”; Q3: “In a typical week, how many days did you eat cooked and/or raw vegetables?”; and Q4: “Usually on the day that you eat cooked and/or raw vegetables, how many servings did you take?”. One serving is defined as the amount of fruit or vegetables eaten for a meal or snack. Respondents were shown photographs of individual amounts (portions) of various FVs for reference to answer the number of daily servings of fruit or vegetables [14].

The average fruit consumption per day was calculated by dividing the product of Q1 and Q2 responses by seven (7), and average vegetable consumption per day was similarly calculated using Q3 and Q4 responses. Adequacy of FV intake was determined according to the Malaysian Dietary Guidelines 2020, in which adequate intake was defined as ≥5 servings per day (i.e., ≥2 servings of fruits and ≥3 servings of vegetables) and insufficient intake was defined as inadequate [14].

#### 2.4.3. Covariates

Several covariates (potential confounders) were included in the regression model, i.e., gender, age, ethnicity, residential area, marital status, educational levels, occupation status, monthly household income, and smoking status. The age of the respondents was categorized as follows: (i) 18–39 years, (ii) 40–59 years, and (iii) 60 years and above. Malaysia is a multiracial country and its ethnicity was classified as follows: (i) Malay, (ii) Chinese, (iii) Indian, and (iv) Others. Residential area was classified as urban and rural, whereas marital status was classified as follows: (i) single, (ii) married, and (iii) widow/widower/divorcee. Based on the local Malaysian education system, we categorized the educational levels of respondents into (i) no formal education, (ii) primary education, (iii) secondary education, and (iv) tertiary education.

Total monthly household income was categorized based on the Malaysian household income classification published by the Department of Statistics Malaysia: (i) bottom 40% (B40) with monthly income below RM 4850 (below USD 1035), (ii) middle 40% (M40) with monthly income between RM 4850 and RM 10,959 (between USD 1035–2338), and (iii) top 20% (T20) with a monthly income of RM 10,960 and above (above USD 2338). The total monthly household incomes are henceforth reported in USD, and values in USD were based on the currency exchange rate on 21 September 2023. Smoking status was categorized as (i) never and (ii) ever (inclusive of both current and past smokers).

We applied the Asia–Pacific classification criteria for both general obesity and central obesity. For general obesity, respondents’ body mass index (BMI) was calculated as the body weight (kg) divided by the square of height (m^2^). Respondents with a BMI of 23 and above were classified as obese, whereas the classification of abdominal obesity was having a waist circumference of 90 cm and above for men, and 80 cm and above for women [17,18].

### 2.5. Statistical Analyses

The statistical analyses were performed using R language. Descriptive statistics and multivariable logistic regression analyses for complex samples were performed utilizing R packages, i.e., the “tidyverse”, “labelled”, “summarytools”, “tibble”, “survey”, “gtsummary” and “flextable” packages in RStudio version 4.3.1 [19]. Furthermore, the percent of correct classification and ROC curve analysis were used to assess the predictive ability of the model. To begin with, a survey design object was produced by incorporating the sampling weights, which were then used in the analyses. A descriptive analysis was performed to determine the proportion of the sociodemographic characteristics (gender, age, ethnicity, residential area, marital status, education level, occupational status, monthly household income) and lifestyle risk factors (i.e., smoking). Pearson’s Chi-Square was used to test the association between sociodemographic characteristics, lifestyle risk factors, and awareness of MHP. A bar chart was used to present the prevalence with a 95% confidence interval (CI) of (i) non-communicable diseases (i.e., general obesity, abdominal obesity, diabetes mellitus, hypertension, and hypercholesterolemia), (ii) MHP awareness, and (iii) adequate FV intake. Complex-sampling descriptive statistics were performed to describe the sociodemographic characteristics and lifestyle factors of respondents, stratified by awareness of MHP. To examine the relationship between awareness of MHP and FV intake, complex-sample multivariable logistic regression was performed, and adjusted odds ratios (aORs) for the associations with their respective 95% confidence intervals (95% CIs) were obtained. A *p*-value less than 0.05 was taken to signify statistically significant associations. The figures were generated using GraphPad Prism version 5.01 for Windows (GraphPad Software, Boston, MA, USA) and the “forestplot” and “readxl” packages in RStudio.

## 3. Results

The sampled population comprised approximately equal proportions of males (50.2%) and females (49.8%) (Table 1). A majority belonged to the 18–39 age group (52.3%), while individuals aged 60 and above constituted 16.3% of the respondents. In terms of ethnic composition, Malays were the largest ethnic group at 56.7%, followed by Chinese (24.0%). A substantial 78.5% resided in urban areas. The majority were married (63.6%) and had obtained secondary education (68.7%). In terms of occupation, 61.8% reported being employed. The B40 represented the largest segment at 62.9%. A large proportion of respondents had never smoked (74.3%).

Significant gender disparity was observed in MHP awareness, only 15.5% of males were aware of MHP compared to 30.5% of females (*p* < 0.001) (Table 2). Awareness of MHP decreased with age; the oldest age group (≥60 years) had the lowest percentage of awareness at 11.8%. Among the ethnic groups, Malays had the highest awareness at 26.3%, while less than 25% of Chinese, Indian, and other ethnicities were aware of MHP (*p* < 0.001). Residents of urban areas were more aware of MHP (24.1%) than in rural areas (18.9%). There were also significant differences across marital status categories and monthly household income brackets. Individuals with tertiary education showed a markedly higher awareness of MHP (42.3%) compared to the other education levels (*p* < 0.001). As for occupation status, respondents who were employed had a slightly higher awareness (*p* = 0.01), whereas never-smokers were more aware of MHP (25.8%) than ever-smokers (15.0%) (*p* < 0.001).

A majority of the respondents had general obesity, with a prevalence of 67.1% (Figure 2, Appendix A). Abdominal obesity was prevalent in 50.4% of the sample. As for non-communicable diseases, diabetes mellitus was identified in 10.3% of the population, making it the third most common disease. Hypertension and hypercholesterolemia were reported by 17.4% and 14.8%, respectively.

The prevalence of MHP awareness was higher in individuals with general obesity (24.9%) and abdominal obesity (25.7%) as compared to those who were non-obese (20.7% and 20.0%, respectively) (Figure 3a and Appendix A). Following this, we observed that the prevalence of MHP awareness was lower in individuals diagnosed with hypertension (20.0%) than in those who were non-diagnosed (23.0%). Meanwhile, the prevalences of MHP awareness were comparable between individuals who were diagnosed with diabetes mellitus (22.9%) and hypercholesterolemia (22.9%) compared to those who did not have these diseases (22.4% and 22.4%, respectively).

An alarmingly low prevalence of adequate FV intake ranged between 2.3% and 2.8% in individuals with general obesity, abdominal obesity, diabetes mellitus, hypertension, and hypercholesterolemia (Figure 3b and Appendix A). We further observed a higher prevalence of adequate FV intake among individuals with hypertension (2.6%) and hypercholesterolemia (2.8%) as compared to those who were not diagnosed with these diseases (2.3% and 2.3%, respectively).

In the general population, the awareness of MHP yielded a statistically non-significant association (odds ratio (aOR): 1.44, 95% CI: 0.85–2.43; *p* = 0.17) when sociodemographic characteristics and lifestyle factors were held constant (Appendix A). However, disease-specific analyses revealed a range of associations. Those with abdominal obesity and awareness of MHP had an increased odds (aOR = 1.86; 95% CI: 1.05–3.29; *p* = 0.03) of adequate FV intake (Figure 4, Appendix A). Notably, individuals diagnosed with diabetes mellitus and MHP awareness demonstrated a marked increased odds (aOR = 6.88; 95% CI: 2.13–22.18; *p* < 0.01) (Figure 4, Appendix A). Similar strong associations were observed for hypertension (aOR = 4.39; 95% CI: 1.96–9.83; *p* < 0.001) and hypercholesterolemia (aOR = 4.16; 95% CI: 1.48–11.72; *p* < 0.01). However, no significant association was observed in those with general obesity (aOR = 1.59; 95% CI: 0.91–2.75; *p* = 0.1).

## 4. Discussion

This is the first study to report the relationship between MHP awareness and adequate FV intake among Malaysian adults with abdominal obesity, diabetes mellitus, hypertension, and hypercholesterolemia using the nationwide representative health and morbidity survey data, namely the NHMS 2019. A significant positive association was observed in this study between awareness of the MHP concept and adequate FV intake among Malaysian adults with abdominal obesity, but not with general obesity. We further observed that Malaysian adults with diabetes mellitus, hypertension, and hypercholesterolemia who are aware of the MHP concept are more likely to have adequate FV intake. The associations were robust and remained significant even after adjusting for demographic characteristics and lifestyle factors.

In our present study, the prevalence of MHP awareness among Malaysian adults with obesity, diabetes, hypertension, and hypercholesterolemia was relatively low and comparable with the reported prevalence of MHP awareness in the general Malaysian population (i.e., 20.4%). The low prevalence of awareness could be attributed to factors such as gender, ethnicity, and occupation status [13]. Further analysis of the nationally representative subset of rural Malaysian adults revealed a significant association between low MHP concept awareness and factors such as being male, middle-aged, having a lower education level, working in the private sector, being retired or unemployed, self-employment, and being an unpaid worker [20]. However, the prevalence of MHP awareness among those with chronic diseases (except for hypertension) was slightly higher, ranging between 22.9% and 25.7%, than the reported prevalence of MHP awareness in the general Malaysian population (20.4%). A plausible explanation is that people with chronic diseases typically have more frequent interactions with healthcare providers. During these interactions, they may have received information and recommendations on their health condition management such as dietary guidance (e.g., the MHP concept). Frequent healthcare facility visits and engagement with healthcare providers can lead to greater awareness of the MHP concept [21]. Furthermore, individuals diagnosed with chronic diseases often become more proactive in seeking information related to dietary choices to manage their health condition, which may include the MHP concept. Hence, they may have greater awareness and be more motivated to adhere to dietary concepts [22].

Although the MHP campaign is a nationwide effort made by the MOH Malaysia, the alarmingly low prevalence of MHP awareness suggests the need to evaluate the effectiveness of the existing strategies in future studies. The consideration of using the latest approach of social media marketing as a promising and strategic tool to advertise the MHP concept may accelerate the efforts to reach out to the public (who are mainly digital users) and subsequently increase the awareness of the MHP concept among the users. Social media platforms, such as TikTok, Facebook, Instagram, YouTube, Twitter, and Threads, have no boundaries and could reach multiple layers of consumers regardless of their residential area, gender, age, and socioeconomic status.

Dietary habits are established at a young age and persist throughout later life, often exhibiting continuity over time [23,24]. Therefore, school health programs, including nutritional education like the concept of MHP and the food pyramid, as well as nutritional rehabilitation programs such as food supplementation, should be further enhanced among primary and secondary school students [25]. Early exposure to the MHP concept during childhood and adolescence may foster the adoption of healthy and balanced dietary habits in adulthood, mitigating the risk of becoming overweight/obese [26] and the development of chronic diseases in the future [27].

Likewise, an alarmingly low prevalence of adequate FV intake among Malaysian adults with obesity, diabetes mellitus, hypertension, and hypercholesterolemia was observed in this study, indicating non-compliance of most of the patients with healthy eating patterns despite being clinically diagnosed. Our study revealed that older age and unemployment were closely associated with low consumption of FV among individuals with abdominal obesity and diabetes mellitus. Meanwhile, among individuals with hypertension and hypercholesterolemia, individuals of Chinese ethnicity were more likely to consume an adequate amount of FVs compared to individuals of other races. There are several possible explanations for these observations. First, older and unemployed individuals may have limited financial resources, making it harder to afford fresh FVs and other nutrient-dense food, which can sometimes be more expensive than processed or convenience foods. A local study that was conducted in the Federal Territory of Kuala Lumpur reported that 89.5% of Malaysian adults in the lower- to medium-income group did not consume an adequate amount of FVs, with price (79.4%) being the top factor affecting food purchase choices [28]. However, we did not find any significant association between household income level and adequate consumption of FV. Second, different cultural habits could potentially influence dietary consumption patterns among the various ethnic groups [29], which were not examined in the present study. Future studies should consider the cultural differences that might affect food choice preferences and food-related beliefs among the multiracial Malaysian population. Third, dining outside the home has become a widespread practice in Malaysia, driven by the growing prosperity, urbanization, evolving lifestyles, and the rising number of working mothers.

Approximately 84.0% of urban Malaysian adults reported eating out for at least one meal per day, two or three days a week [30]. However, this trend comes at the expense of an increasing prevalence of non-communicable diseases (NCDs) and their associated risk factors. Previous studies have demonstrated that eating outside the home, combined with the shift in dietary consumption patterns from a traditional to a more Westernized diet, is associated with poor dietary quality. This is characterized by high levels of saturated fat, sugar, and salt, as well as a deficiency in dietary fiber and essential micronutrients [31,32,33]. The Seattle Obesity Study, a cross-sectional study comprising 2001 adult male and female residents of King County, Washington, reported that a higher consumption frequency of eating out (i.e., foods away from home) was linked to lower FV intake among adults [34]. The authors of the study suggested that the absence of a strong demand for healthier food options has led to a limited availability of such choices in food outlets and restaurants. Furthermore, healthier menu items often have a shorter shelf life, require longer preparation time, have lower sales, and have higher labor costs. These factors may explain why individuals who frequently dine out tend to consume fewer fruits and vegetables.

The positive association observed between MHP awareness and adequate FV intake among Malaysian adults with abdominal obesity, diabetes mellitus, hypertension, and hypercholesterolemia in the present study indicated that those who are aware of the MHP concept have a better chance of having adequate FV intake. In a cross-sectional study assessing the promotion, awareness, and preferences of dietary guidelines among Saudi adults, it was reported that approximately 15% of the respondents were aware of the Saudi Healthy Eating Plate, and roughly 4.5% consumed five portions of fruits and vegetables daily. However, the study did not further investigate the relationship between awareness of the healthy eating plate and FV intake [35]. Another cross-sectional survey, along with in-depth interviews, aimed to assess the factors associated with meeting the healthy plate recommendation. The study reported that close to three out of every five primary care patients in Singapore with prediabetes follow the healthy plate recommendation [36].

In our study, we did not investigate the relationship between the MHP practice and adequate FV intake due to data limitations. However, it is believed that individuals with chronic diseases such as obesity, diabetes, hypertension, and dyslipidemia are more likely to comply with non-pharmacological management strategies, such as dietary modifications that emphasize the importance of adequate FV intake for the management of their health conditions [37,38,39]. Awareness of MHP may assist them in making healthier food choices with appropriate portions and thus improve their adherence to consuming at least five servings of FVs daily [22]. Furthermore, there is an urgent need for the identification of potential barriers to the promotion of the MHP concept, especially when targeting those with NCDs. This can assist healthcare professionals in developing interventions customized to address these issues, ultimately leading to greater awareness of the MHP concept and, consequently, improved FV intake among morbid individuals [40].

## 5. Strengths, Limitations, and Future Works

This present study utilized the data from the nationwide health and morbidity survey with a large number of respondents who are Malaysian citizens, allowing the findings to be generalized to the non-institutionalized population of Malaysia. Furthermore, the important potential confounders, including sociodemographic characteristics and lifestyle risk factors, which might confound the association between MHP awareness and adequate FV intake, were controlled during the association analysis.

Several limitations in the present study were acknowledged. To begin with, there was a potential self-reporting bias that could arise from the recall period/recall bias for adequacy in FV intake, where respondents were asked to recall the intake of FVs for the past week. In addition to this potential bias, the present study is limited to accessing the relationship of MHP awareness with adequate FV intake, but not the MHP knowledge and MHP practice with adequate FV intake. Hence, no further analyses could be performed to investigate the relationship between MHP knowledge, MHP practice, and adequate FV intake in order to gain better insights into the efficiency of the MHP campaign.

Having said this, a future quantitative study about the knowledge, attitude, and practice of the MHP concept among Malaysian adults with NCDs is needed to provide more useful information to identify the knowledge gaps and behavioral patterns, as well as to investigate the correlation between MHP practice and adequate FV intake. These kinds of findings could aid public health authorities and clinicians in accessing the respective efficacy of the MHP campaign and compliant non-pharmacological management (dietary modification) among Malaysian adults with NCDs.

## 6. Conclusions

In conclusion, we observed that Malaysians with NCDs (i.e., abdominal obesity, diabetes mellitus, hypertension, and hypercholesterolemia) who are aware of the MHP concept are more likely to have an adequate FV intake. This indicates the need for ongoing efforts by policymakers, healthcare professionals, and nutrition educators to promote the concept of MHP that will ensure sufficient intake of FVs or adoption of a healthy eating pattern to achieve and maintain optimal health among Malaysian adults with NCDs.

## Figures and Tables

**Figure 1 nutrients-15-05043-f001:**
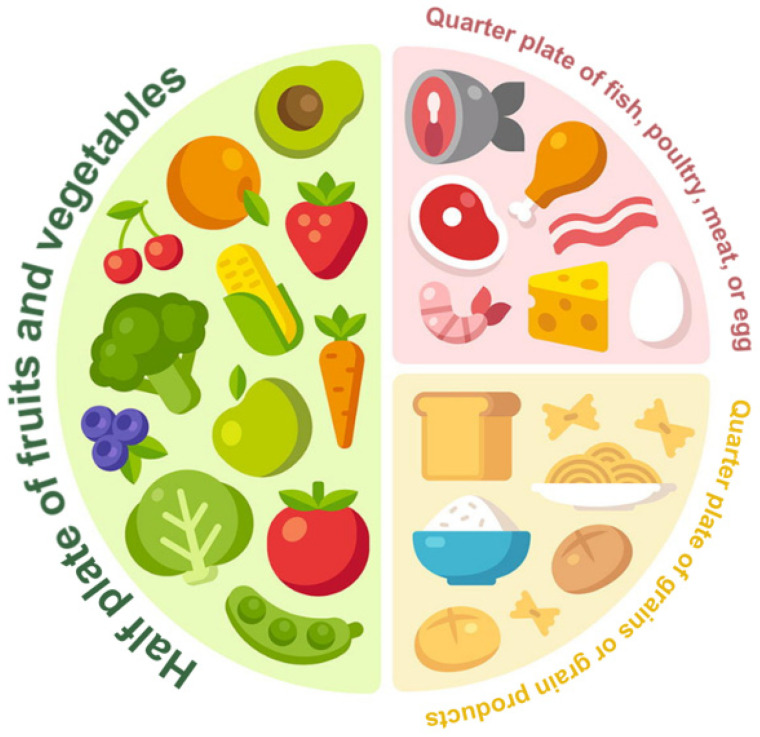
The Malaysian Healthy Plate (adapted from Malaysian Dietary Guidelines 2020 [2] and online image (1.5)).

**Figure 2 nutrients-15-05043-f002:**
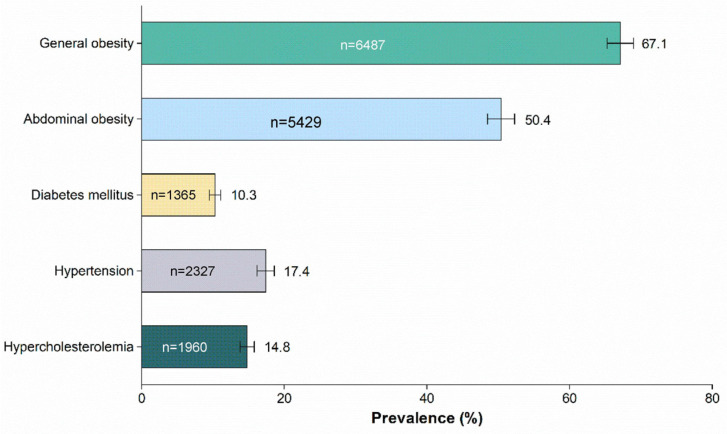
Prevalence of general obesity, abdominal obesity, diabetes mellitus, hypertension, and hypercholesterolemia among Malaysian adults aged 18 years and above (*n* = 9760).

**Figure 3 nutrients-15-05043-f003:**
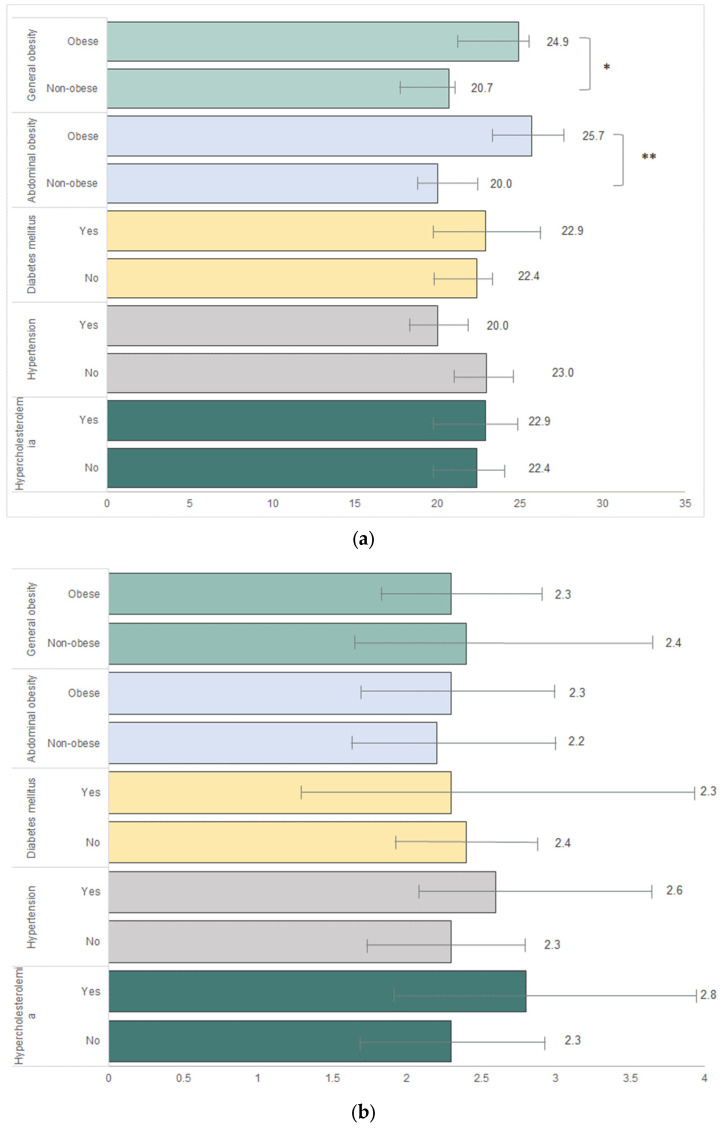
The prevalence of MHP awareness and adequate FV intake among Malaysian adults aged 18 years and above. (**a**) The prevalence of MHP awareness among Malaysian adults with general obesity, abdominal obesity, diabetes mellitus, hypertension, and hypercholesterolemia. (**b**) The prevalence of adequate FV intake among Malaysian adults with general obesity, abdominal obesity, diabetes mellitus, hypertension, and hypercholesterolemia. * = *p* < 0.01; ** = *p* < 0.001.

**Figure 4 nutrients-15-05043-f004:**
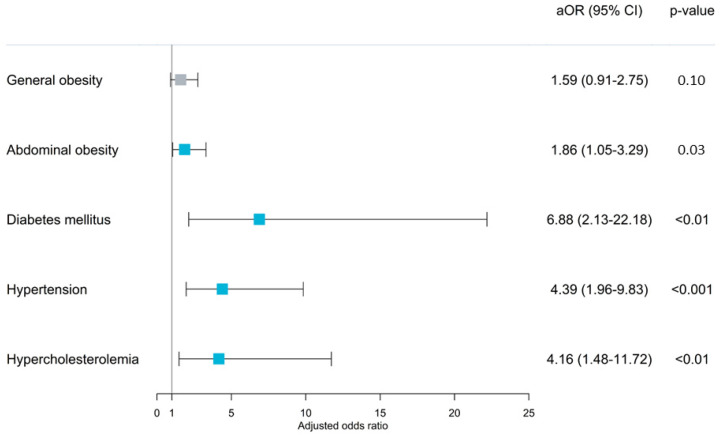
Associations between MHP awareness and adequate FV intake among Malaysian adults aged 18 years and above with general obesity, abdominal obesity, diabetes mellitus, hypertension, and hypercholesterolemia. Multiple logistic regression was performed and adjusted for sociodemographic (i.e., sex, age, ethnicity, residential area, marital status, education, and monthly income) and lifestyle risk factors (i.e., smoking status). There were no significant interactions among the independent variables. aOR: adjusted odds ratio; 95% CI: 95% confidence interval.

**Table 1 nutrients-15-05043-t001:** Characteristics of NHMS 2019 respondents, Malaysian adults aged 18 years and above with morbidities (*n* = 9760).

Characteristic	Estimated Population	Count (*n*)	%	95% CI ^1^
** *Sociodemographic* **				
Gender				
Male	9,430,670	4427	50.2	49.0–51.4
Female	9,367,092	5333	49.8	48.6–51.0
*Age group (years old)*				
18–39	9,828,210	3894	52.3	50.7–53.8
40–59	5,900,808	3465	31.4	30.0–32.9
≥60	3,068,744	2401	16.3	15.1–17.6
*Ethnicity*				
Malay	10,656,046	6655	56.7	52.4–60.9
Chinese	4,510,234	1315	24.0	20.1–28.3
Indian	1,174,130	639	6.2	5.0–7.8
Others	2,457,353	1151	13.1	11.0–15.4
*Residential area*				
Urban	14,753,184	5908	78.5	77.0–79.9
Rural	4,044,578	3852	21.5	20.1–23.0
*Marital status*				
Single	5,455,290	2071	29.0	27.3–30.8
Married	11,950,450	6619	63.6	61.6–65.5
Widow/Widower/Divorcee	1,392,022	1070	7.4	6.7–8.2
*Education level*				
No formal education	713,115	530	3.8	3.2–4.4
Primary education	3,111,533	2113	16.6	15.4–17.8
Secondary education	12,901,427	6218	68.6	67.0–70.3
Tertiary education	2,058,279	891	11.0	9.7–12.4
*Occupation status*				
Employed	11,618,678	5464	61.8	60.1–63.5
Unemployed	7,177,953	4295	38.2	36.5–39.9
*Monthly household income*				
Bottom 40%	11,117,530	6193	62.9	60.1–65.7
Middle 40%	4,842,580	2190	27.5	25.0–29.9
Top 20%	1,701,166	765	9.6	8.0–11.5
** *Lifestyle* **				
*Smoking*				
Ever-smoker	4,839,808	2340	25.7	24.3–27.3
Never-smoker	13,957,953	7420	74.3	72.7–75.7

^1^ 95% CI: 95% confidence interval.

**Table 2 nutrients-15-05043-t002:** Awareness of Malaysian Healthy Plate by sociodemographic factors among NHMS 2019 respondents, Malaysian adults aged 18 years and above with morbidities (*n* = 9760).

Characteristic	Estimated Population	Count (*n*)	Not Aware (*n* = 7201)		Aware (*n* = 2295)		*p*-Value
			Prevalence (%)	95% CI ^1^	Prevalence (%)	95% CI ^1^	
** *Sociodemographic* **							
*Gender*							<0.001
Male	9,430,670	4427	84.5	82.7–86.1	15.5	13.9–17.3	
Female	9,367,092	5333	69.5	67.2–71.7	30.5	28.3–32.8	
*Age group (years old)*							<0.001
18–39	9,828,210	3894	73.8	71.4–76.0	26.2	24.0–28.6	
40–59	5,900,808	3465	76.7	74.3–78.9	23.3	21.1–25.7	
≥60	3,068,744	2401	88.2	85.9–90.1	11.8	9.9–14.1	
*Ethnicity*							<0.001
Malay	10,656,046	6655	73.7	71.8–75.6	26.3	24.4–28.2	
Chinese	4,510,234	1315	82.2	78.3–85.5	17.8	14.5–21.7	
Indian	1,174,130	639	77.0	71.4–81.7	23.0	18.3–28.6	
Others	2,457,353	1151	81.9	78.2–85.1	18.1	14.9–21.8	
*Residential area*							<0.001
Urban	14,753,184	5908	75.9	73.9–77.8	24.1	22.2–26.1	
Rural	4,044,578	3852	81.1	78.8–83.2	18.9	16.8–21.2	
*Marital status*							<0.001
Single	5,455,290	2071	79.4	76.7–81.8	20.6	18.2–23.3	
Married	11,950,450	6619	75.1	73.1–76.9	24.9	23.1–26.9	
Widow/Widower/Divorcee	1,392,022	1070	84.9	81.6–87.8	15.1	12.2–18.4	
*Education level*							<0.001
No formal education	713,115	530	93.8	89.7–96.3	6.2	3.7–10.3	
Primary education	3,111,533	2113	88.7	86.2–90.8	11.3	9.2–13.8	
Secondary education	12,901,427	6218	76.4	74.5–78.2	23.6	21.8–25.5	
Tertiary education	2,058,279	891	57.7	52.4–62.7	42.3	37.3–47.6	
*Occupation status*							0.01
Employed	11,618,678	5464	75.7	73.6–77.6	24.3	22.4–26.4	
Unemployed	7,177,953	4295	79.2	76.9–81.3	20.8	18.7–23.1	
*Monthly household income*							<0.001
Bottom 40%	11,117,530	6193	79.0	77.2–80.7	21.0	19.3–22.8	
Middle 40%	4,842,580	2190	75.0	71.6–78.1	25.0	21.9–28.4	
Top 20%	1,701,166	765	68.9	63.6–73.8	31.1	26.2–36.4	
** *Lifestyle* **							
*Smoking*							<0.001
Ever-smoker	4,839,808	2340	85.0	82.4–87.2	15.0	12.8–17.6	
Never-smoker	13,957,953	7420	74.2	72.3–76.1	25.8	23.9–27.7	

^1^ 95% CI: 95% confidence interval.

## Data Availability

All the generated data during this study are included in this published article and its Appendix A. For data protection purposes, the data used for this study are not publicly available but are available from the Sector for Biostatistics and Data Repository, Office of NIH Manager, National Institutes of Health Malaysia upon reasonable request and with permission from the Director General of Ministry of Health Malaysia.

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
