# Peer review of "Does Awareness of Malaysian Healthy Plate Associate with Adequate Fruit and Vegetable Intake among Malaysian Adults with Non-Communicable Diseases?"

_nutrients, 2023, doi:10.3390/nu15245043_

Round 1

Reviewer 1 Report

Comments and Suggestions for Authors

The main research question, according to the authors, was the relationship between Malaysian Healthy Plate (MHP) and adequate fruit and vegetable (FV) intake among the morbid Malaysian adults.

The topic of the publication is investigating the relationship between awareness of the MHP and FV intake among the morbid Malaysian adults.

This study adds to the scientific knowledge on the association between awareness of MHP and FV intake among Malaysian adults with obesity, diabetes, hypertension and hypercholesterolemia.

The methodology used by the authors was well selected and described. the association between awareness of MHP intake and FV among Malaysian adults with obesity, diabetes, hypertension and hypercholesterolemia was determined using multivariate logistic regression controlling for sociodemographic characteristics and lifestyle risk factors.

The conclusions are consistent with the description of the results and the presentation of the results in the form graphs.

The literature presented is appropriately selected for the research topic, even though the number of publications directly related to the topic is limited. Nevertheless, the authors have made adequate reference to the available literature sources.

Chartes and figures are correctly prepared and add value to the content of the manuscript. 

The results highlight the need for continuous efforts by policymakers, health care professionals and nutrition educators to promote the concept of MHP and ensure that sick Malaysian adults consume enough FV or adopt a healthy dietary pattern to achieve and maintain optimal health.

It is worth considering the development of long-term programs that could be implemented by various institutions and have a measurable impact on improving society's awareness of healthy eating.

The methodology used by the authors was well selected and described. 

The conclusions are consistent with the description of the results and the presentation of the results in the form graphs.

The literature presented is appropriately selected for the research topic, even though the number of publications directly related to the topic is limited. Nevertheless, the authors have made adequate reference to the available literature sources.

Chartes and figures are correctly prepared and add value to the content of the manuscript. 

Author Response

For research article: Does awareness of Malaysian Healthy Plate associates with adequate fruit and vegetable intake among Malaysian adults with non-communicable diseases? [Manuscript ID: nutrients-2717067]

Responses to Reviewer 1’s Comments

  1. Summary

On behalf of all the co-authors, thank you very much for taking time to review this manuscript. Please find the detailed responses below and the corresponding revisions/corrections highlighted/in track changes in the re-submitted files

  1. Questions for general evaluation

Questions for General Evaluation

Reviewer’s Evaluation

Response and Revisions

Does the introduction provide sufficient background and include all relevant references?

Yes

Corresponding response in the point-by-point response as below

Are all the cited references relevant to the research?

Yes

Corresponding response in the point-by-point response as below

Is the research design appropriate?

Yes

Corresponding response in the point-by-point response as below

Are the methods adequately described?

Yes

Corresponding response in the point-by-point response as below

Are the results clearly presented?

Yes

Corresponding response in the point-by-point response as below

Are the conclusions supported by the results?

Yes

Corresponding response in the point-by-point response as below

  1. Point-by-point response to comments and suggestions for authors

Point 1: The main research question, according to the authors, was the relationship between Malaysian Healthy Plate (MHP) and adequate fruit and vegetable (FV) intake among the morbid Malaysian adults.

Authors’ response: The authors thank the reviewer for his/her conclusion from the review

Point 2: The topic of the publication is investigating the relationship between awareness of the MHP and FV intake among the morbid Malaysian adults.

Authors’ response: The authors thank the reviewer for his/her conclusion from the review

Point 3: This study adds to the scientific knowledge on the association between awareness of MHP and FV intake among Malaysian adults with obesity, diabetes, hypertension and hypercholesterolemia.

Authors’ response: The authors thank the reviewer for his/her conclusion from the review

Point 4: The methodology used by the authors was well selected and described. the association between awareness of MHP intake and FV among Malaysian adults with obesity, diabetes, hypertension and hypercholesterolemia was determined using multivariate logistic regression controlling for sociodemographic characteristics and lifestyle risk factors.

Authors’ response: The authors thank the reviewer for his/her conclusion from the review. The authors also would like to thank the reviewer for his/her compliment in this piece of work

Point 5: The conclusions are consistent with the description of the results and the presentation of the results in the form graphs.

Authors’ response: The authors thank the reviewer for his/her conclusion from the review. The authors also would like to thank the reviewer for his/her compliment in this piece of work

Point 6: The literature presented is appropriately selected for the research topic, even though the number of publications directly related to the topic is limited. Nevertheless, the authors have made adequate reference to the available literature sources.

Authors’ response: The authors thank the reviewer for his/her conclusion from the review. The authors also would like to thank the reviewer for his/her compliment in this piece of work

Point 7: Charts and figures are correctly prepared and add value to the content of the manuscript.

Authors’ response: The authors thank the reviewer for his/her conclusion from the review. The authors also would like to thank the reviewer for his/her compliment in this piece of work

Point 8: The results highlight the need for continuous efforts by policymakers, health care professionals and nutrition educators to promote the concept of MHP and ensure that sick Malaysian adults consume enough FV or adopt a healthy dietary pattern to achieve and maintain optimal health.

Authors’ response: The authors thank the reviewer for his/her conclusion from the review. The authors also would like to thank the reviewer for his/her compliment in this piece of work

Point 9: It is worth considering the development of long-term programs that could be implemented by various institutions and have a measurable impact on improving society's awareness of healthy eating.

Authors’ response: The authors thank the reviewer for his/her conclusion from the review. The authors also would like to thank the reviewer for his/her compliment in this piece of work

Point 10: The conclusions are consistent with the description of the results and the presentation of the results in the form graphs.

Authors’ response: The authors thank the reviewer for his/her conclusion from the review. The authors also would like to thank the reviewer for his/her compliment in this piece of work

Reviewer 2 Report

Comments and Suggestions for Authors

The manuscript investigated the association of awareness of the MHP and FV intake among the morbid Malaysian adults. The study has a large sample size and is highly relevant and instructive, but there are still the following areas that need to be revised.

1. Line 24, line 144, line 224 and line 236, the sample sizes described above are inconsistent.

2. Study design and sampling section, a detailed inclusion exclusion process, including screening of covariate data, should be demonstrated in the manuscript and draw an appropriate flow chart.

3. Line 174, Please explain what “serving” means.

4. Lines 184-186, alcohol assumption and physical activity are also potential confounding factors. If possible, it is recommended to include them as covariates in logistic regression.

5. Statistical analysis section, the statistical description of the variables and the method of testing for between-group variability were not mentioned.

6. Lines 200-201, it is inappropriate to categorize current smokers and past smokers as ever smokers, suggesting that smoking be categorized as smokers (current and past) and never smokers.

7. The proportions of the sections in the table 1 do not add up to 100% and should be double checked by the authors.

8. The contents of lines 224-226 are not in Table 1. Throughout the results section, it is recommended that the author match the tables to the corresponding descriptions.

9. Lines 259-266, lines 281-183, without a test of variance, the results described cannot be obtained based on the tables.

10. The tables in the article are not standard three-line tables. The number of decimal places of p figure4 is not uniform.

Author Response

For research article: Does awareness of Malaysian Healthy Plate associates with adequate fruit and vegetable intake among Malaysian adults with non-communicable diseases? [Manuscript ID: nutrients-2717067]

Responses to Reviewer 2’s Comments

  1. Summary

On behalf of all the co-authors, thank you very much for taking time to review this manuscript. Please find the detailed responses below and the corresponding revisions/corrections highlighted/in track changes in the re-submitted files

  1. Questions for general evaluation

Questions for General Evaluation

Reviewer’s Evaluation

Response and Revisions

Does the introduction provide sufficient background and include all relevant references?

Yes

Are all the cited references relevant to the research?

Yes

Is the research design appropriate?

Yes

Are the methods adequately described?

Must be improved

The methods section is revised and improvement has been made accordingly

Are the results clearly presented?

Can be improved

The results are reviewed and revised for clearer presentation

Are the conclusions supported by the results?

Yes

  1. Point-by-point response to comments and suggestions for authors

Point 1: Line 24, line 144, line 224 and line 236, the sample sizes described above are inconsistent.

Author’s response: We have double-checked the sample size number throughout the manuscript where data from a total of 9,760 respondents were included in the descriptive analysis. The subsequent logistic regression analysis, the data from a total of 8,164 was included. We have revised the sentence on line 161-164 in the revised manuscript to make the description of the sample size clearer.

Point 2: Study design and sampling section, a detailed inclusion exclusion process, including screening of covariate data, should be demonstrated in the manuscript and draw an appropriate flow chart.

Author’s response: The inclusion and exclusion process has been detailed and can be found from line 154 to line 164 in the revised version of manuscript. We agreed with the suggestion of reviewer to include the flow chart and has been included in the supplementary file (Figure S1) as per reviewer’s comment.

Point 3: Line 174, Please explain what “serving” means.

Author’s response: We have defined the word “serving” in line 195 -196 in the revised version manuscript.

Point 4: Lines 184-186, alcohol assumption and physical activity are also potential confounding factors. If possible, it is recommended to include them as covariates in logistic regression.

Author’s response: We acknowledge that the alcohol consumption and physical activity are one of the independent risk factors for non-communicable diseases but not Malaysian Healthy Plate awareness (reference number 20). Hence, it was not included in the final model.

Point 5: Statistical analysis section, the statistical description of the variables and the method of testing for between-group variability were not mentioned.

Author’s response: We have included the statistical description from line 236 to line 244.

Point 6: Lines 200-201, it is inappropriate to categorize current smokers and past smokers as ever smokers, suggesting that smoking be categorized as smokers (current and past) and never smokers.

Author’s response: Smoking is included in the final model as control variable and it is not the variable of interest in this present study.

Point 7: The proportions of the sections in the table 1 do not add up to 100% and should be double checked by the authors.

Author’s response: We have double-checked and make the correction accordingly.

Point 8: The contents of lines 224-226 are not in Table 1. Throughout the results section, it is recommended that the author match the tables to the corresponding descriptions.

Author’s response: We have revised the result section and matched Table 1 to the corresponding descriptions from line 254 to line 261

Point 9: Lines 259-266, lines 281-283, without a test of variance, the results described cannot be obtained based on the tables.

Author’s response: We have performed the test of variance and included the results in the revised the figure 3a and 3b

  1. The tables in the article are not standard three-line tables. The number of decimal places of p figure4 is not uniform.

Author’s response: We have revised the tables’ format and the number of decimal places of the p-value in Figure 4 have been standardized to three decimal point.

Reviewer 3 Report

Comments and Suggestions for Authors

Does awareness of Malaysian Healthy Plate associates with adequate fruit and vegetable intake among Malaysian adults with non-communicable diseases?

Dear authors I want to congratulate with your paper, but I have some considerations to do

The main question is “awareness of Malaysian Healthy Plate associates with adequate fruit and vegetable intake among Malaysian adults with 3 non-communicable diseases

The topic is more or less original and the authors try to help with their results the public health authorities. to reinforce the existing strategies to increase and promote awareness and adherence to the MHP practice (behavioral intervention) that focus on high quality, healthy and balanced diet which is of paramount importance in the management of morbid Malaysian adults.

This article does not add anything new to this subject area due to the authors focused their study to promote balanced diet to manage morbid Malaysian adults, as previous studies.

It is not really clear the Methodology, I recommended the authors to improve the study design.

The conclusions are consistent and appropriate.

The references are appropriate, but some are cited twice, and the author have to remove the duplicates.

1.     you have to change (citation) by [ this ]

2.     I recommended the authors to improve the Introduction, including more recently studies around the world

3.     Cite 17 and 18 is duplicated, you have to remove one.

4.     Cite 14 must be revised.

5.     Cite 6 must be completed.

6.     Cite 9 must be revised totally.

In my opinion, the tables and figures are properly presented.

Author Response

For research article: Does awareness of Malaysian Healthy Plate associates with adequate fruit and vegetable intake among Malaysian adults with non-communicable diseases? [Manuscript ID: nutrients-2717067]

Responses to Reviewer 3’s Comments

  1. Summary

On behalf of all the co-authors, thank you very much for taking time to review this manuscript. Please find the detailed responses below and the corresponding revisions/corrections highlighted/in track changes in the re-submitted files

  1. Questions for general evaluation

Questions for General Evaluation

Reviewer’s Evaluation

Response and Revisions

Does the introduction provide sufficient background and include all relevant references?

Can be improved

Corresponding response in the point-by-point response as below

Are all the cited references relevant to the research?

Can be improved

Corresponding response in the point-by-point response as below

Is the research design appropriate?

Yes

Corresponding response in the point-by-point response as below

Are the methods adequately described?

Yes

Corresponding response in the point-by-point response as below

Are the results clearly presented?

Yes

Corresponding response in the point-by-point response as below

Are the conclusions supported by the results?

  1. Point-by-point response to comments and suggestions for authors

Point 1: It is not really clear the Methodology, I recommended the authors to improve the study design.

Authors’ response: We have revised the study design and sampling section according to the reviewer’s comment from line 127 to line 136 in the revised version of manuscript.

Point 2: The references are appropriate, but some are cited twice, and the author have to remove the duplicates.

Authors’ response: We double-checked the references and removed the duplicates accordingly.

Point 3: you have to change (citation) by [ this ]

Authors’ response: We have revised the format of the citation accordingly.

Point 4: I recommended the authors to improve the Introduction, including more recently studies around the world.

Authors’ response: We have revised and improved the Introduction as per reviewer’s comment, from line 52 to line 66 in the revised version of manuscript.

Point 5: Cite 17 and 18 is duplicated, you have to remove one

Authors’ response: We have doubled-checked and removed the duplicates reference.

Point 6: Cite 14 must be revised.

Authors’ response: We have revised the citation number 14.

Point 7: Cite 6 must be completed

Authors’ response: We have revised the citation number 6.

Point 8: Cite 9 must be revised totally.

Authors’ response: We have revised the citation number 9.

Round 2

Reviewer 3 Report

Comments and Suggestions for Authors

no more comments